# ESS Neutrino Super Beam ESSνSB Design and Performance for Precision Measurements of the Leptonic CP Violating Phase δ$_{CP}$ †

**Tord Ekelöf**  **on behalf of the ESSnuSB Collaboration**

Department of Physics and Astronomy, Uppsala University, 752 36 Uppsala, Sweden; tord.ekelof@physics.uu.se;
Tel.: +46-704250210

† Presented at the 23rd International Workshop on Neutrinos from Accelerators, Salt Lake City, UT, USA, 30–31 July 2022.

**Abstract:** A design study ESSνSB was carried out during the years 2018–2021 concerning how the five MW linear proton accelerators of the European Spallation Source, which are currently under construction in Lund, Sweden, can be used to generate a world-unique, intense neutrino Super Beam for precision measurements of the leptonic CP violating phase δ$_{CP}$. As there are definite limits, which are related to uncertainties in neutrino–nucleus interaction modeling, to how far the systematic errors in such measurements can be reduced, the method chosen in this project is to make the measurements at the second oscillation maximum, where the CP violation signal is close to three times larger than at the first, whereas the systematic errors are approximately the same at the two maxima. As the second maximum is located three times further away from the neutrino source than the first maximum, a higher neutrino beam intensity and thus a higher proton driver power are required when measuring at the second maximum. The unique high power of the ESS proton linac will allow for the measurements to be made at the second maximum and thereby for the most precise measurements of the leptonic CP violation phase δ$_{CP}$ to be made. This paper describes the results of the work made on the conceptual design of ESSνSB layout, infrastructure, and components as well as the evaluation of the physics performance for leptonic CP violation discovery and, in particular, the precision in the measurement of δ$_{CP}$.

**Keywords:** neutrino oscillation; leptonic CP violation phase; second oscillation maximum



## 1. Introduction

The discovery of leptonic CP violation and the measurement of its strength are primary milestones on the road toward understanding why there is matter and no antimatter in the universe. Another outstanding question is whether sterile neutrinos exist, as some experimental results indicate this but do not confirm it with a sufficiently high level of confidence. Generally, high precision measurements of neutrino flavor oscillations have significant potential to shed light on physics beyond the standard model. The value eventually found for the leptonic CP violation phase δ$_{CP}$ will be used to discriminate between different leptogenesis models, explaining the matter–antimatter symmetry in the universe [1] and also between different flavor models [2].

## 2. The Design of the ESS Neutrino Super Beam Project ESSνSB

The European Spallation Source (ESS) research facility [3] will, when it has reached its design performance, be operating with the most powerful accelerator in the world. This implies that it will have world-leading capabilities, not only for material science as a neutron spallation source but also for fundamental particle–physics research, like the precision measurement of neutrino oscillations with the prime aim to search for and measure leptonic CP violations.

A conceptual design of the ESS neutrino Super Beam (ESSνSB) project was produced during 2018–2022 by the ESSνSB consortium, currently consisting of some 70 physicists and engineers at 20 European universities and laboratories, with the support of these institutions and from the European Union Horizon 2020 programme. The results have recently been published in an ESSνSB Conceptual Design Report [4]. This report describes the conceptual design of the ESS accelerator upgrade, of the accumulator ring, of the target station, of the near neutrino detector located at the ESS site, and of the far large neutrino detector located in the Zinkgruvan mine in Sweden, as well as the results of physics performance evaluations of ESSνSB and a first estimate of the investment costs of the facility.

The pi mesons generated in the collision of the ESS proton beam with the neutrino production target will decay, generating a flux of muons and muon neutrinos. Using a van-der-Meer horn to focus the pi mesons before they decay in the decay tunnel located downstream of the target and stopping the muons in a beam stopper at the end of the decay tunnel, a beam of nearly exclusively muon neutrinos will be produced. Due to the interference between the different neutrino mass states, electron neutrinos will appear at a rate with consecutive maxima and minima along the beamline, the distances from the source of which are proportional to the neutrino energy. It is from the rate and energy spectrum of electron neutrinos observed in a neutrino beam and those of electron anti-neutrinos in an anti-neutrino beam, respectively, and their ratio, that the leptonic CP violation signal can be derived. This signal is close to three times higher at the second neutrino oscillation maximum compared to the first. As the neutrino beam is divergent and the second oscillation maximum is located three times further away from the source, a significantly higher neutrino beam intensity and/or larger detector volume are needed to measure at the second oscillation maximum compared to the first. The energy of the neutrinos in the beam is spread over a comparatively wide range. The positions of the oscillation maxima for the individual neutrinos are therefore spread over correspondingly wide ranges along the beamline. The neutrino detectors of existing and other planned long-baseline neutrino experiments are located such that the measurements are predominantly made in the region of the first neutrino oscillation maximum. The strength of ESSνSB is that it is possible, owing to the uniquely high power of the ESS accelerator, to locate the far neutrino detector further down the beam line, making measurements predominantly in the region of the second oscillation maximum. With the 5 MW ESS linear accelerator beam, the intensity of the ESSνSB neutrino beam will, given the 540,000 m$^3$ large volume of the two far water Cherenkov detectors, be sufficiently high enough to allow statistically significant measurements to be made at the second oscillation maximum. Due to the lack of knowledge of the complex interactions of the neutrinos with the target nuclei in the water, the possibilities for discovery and measurement of the leptonic CP violation signal will be limited foremost by systematic rather than statistical errors. As the systematic errors will be approximately the same at the first and second oscillation maximums, having a signal close to a factor three times higher is equivalent to having a relative measurement error that is smaller by approximately the same factor. This implies a significantly higher potential for the discovery of leptonic CP violation and, in particular, higher precision in the measurement of $\delta_{CP}$.

The layout of ESSνSB is shown in Figure 1. For the realization of the ESSνSB project, the beam power of the ESS accelerator will be upgraded from 5 MW to 10 MW by injecting and accelerating 14 extra pulses per second, each containing ca $10^{15}$ H$^-$ ions, interleaved with the 14 proton pulses per second that the accelerator will accelerate and deliver concurrently to the ESS neutron spallation target. The pulse length is ca 3 milliseconds, and by adding the 14 H$^-$ pulses, the accelerator duty cycle will thus be raised from ca 4% to ca 8%. This will require, in addition to a H$^-$ source to be installed at the side of the proton source, a doubling of the power delivered by the accelerator RF power amplifiers as well as the capacity to cool the accelerating cavities. Each of the H$^-$ pulses will be split up into four sub-pulses separated by 0.1 ms gaps using a beam chopper at the upstream end of the

accelerator. After acceleration to full energy, the H$^-$ pulse will be extracted from the linear accelerator into the transfer line and further injected into the accumulator ring, at the entrance of which the H$^-$ ions will be stripped of their electrons and the resulting protons will be injected into the circulating proton beam. Each of the four ca 0.65 ms long sub-pulses will be accumulated in the sequence in the 384 m circumference accumulator ring in ca 2400 turns and then extracted from the ring in one turn, resulting in four consecutive compressed proton pulses, each ca 1.3 microseconds long and containing ca $2.5 \times 10^{14}$ protons. Each of the four sub-pulses will be guided to one of the four parallel transferred lines in the switchyard and sent to the ESSνSB target station, where each sub-pulse will hit one of the fout targets, each designed to sustain a beam power of ca 1.25 MW. The pi mesons produced in the target will be focused on the forward direction by van der Meer horns into the 50 m long decay tunnel, where each pi meson will decay to one muon and one muon neutrino. The muons will be absorbed at the end of the decay tunnel, and the so-formed neutrino beam, directed slightly downwards from the horizontal plane, will continue ca 250 m through the underground to the near detector, where the neutrino beam will be monitored and the neutrino cross-section will be measured. The neutrino beam will continue almost unattenuated for 360 km through the underground to the two large far detectors located at ca 1000 m depth in the Zinkgruvan mine, where the electron and muon neutrinos that interact with the water in the detectors will be identified and their incident energies will be reconstructed. The plan is to run two periods, initially of 5 years each, with opposite polarities of the magnetic field in the van der Meer horn, thereby focusing pi mesons of opposite charges and generating a neutrino and an antineutrino beam, respectively. In this way, a comparison of the muon-to-electron neutrino and the muon-to-electron antineutrino oscillation rates and energy spectra can be made, seeking to detect and measure a difference, till today not yet discovered, in the behavior of leptonic matter and antimatter, which would be a signal of leptonic CP violation.

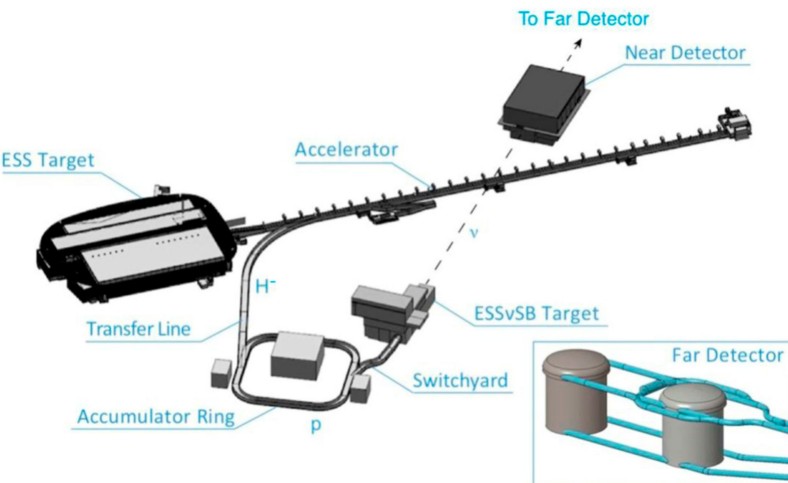

**Figure 1.** Layout of the ESSνSB project showing the ESS proton linac, which will accelerate 14 additional 2.86 ms long H$^-$ ion pulses per second, interleaved with the ordinary proton pulses, which will be transferred to the accumulator ring where they will be stripped of their electrons at the injection point. The resulting proton pulses will be accumulated in the ring and then extracted in one turn, forming 1.3 μs short pulses that will be directed toward the target station, where they will produce a high flux of pi mesons. The neutrino beams resulting from the pi meson decay in the decay tunnel downstream of the target will be monitored by the near detector, and the neutrino flavor oscillations will be measured in the beam by the far detector, located 360 km from ESS.

## 3. The Physics Performance of ESSνSB

The results of the evaluation, based on Monte Carlo simulations, of the potential for CP violation discovery and measurement with ESSνSB, were published in 2022 [4] and are

shown in Figure 2. After 10 years of data collection with ESSνSB, leptonic CP violation will be discovered in over 70% of the range of possible $\delta_{CP}$ values, and the error in the determination of the $\delta_{CP}$ phase angle will be smaller than 8° for all possible values of $\delta_{CP}$. In 2022, the performance of the future long-baseline experiments of DUNE [5] and Hyper-K was also published [6]. A comparison of these published results confirms the superior performance of ESSνSB, which is due to its close to three times higher sensitivity to the CP violation effect at the second oscillation maximum.

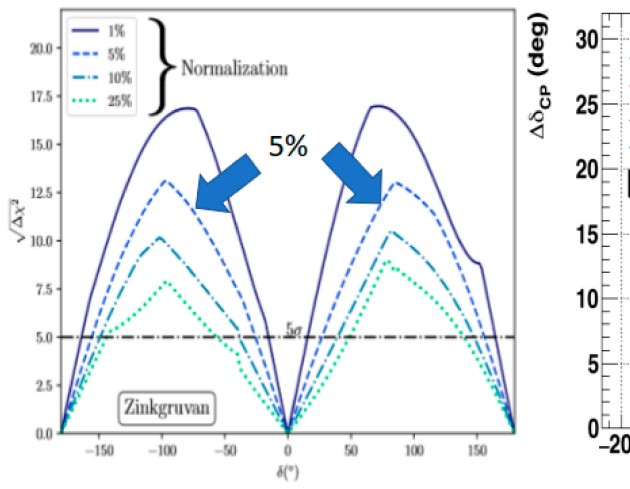

Discovery potential vs $\delta_{CP}$ angle after 10 years with 5% normalization error providing 70% coverage of all $\delta_{CP}$ vaues

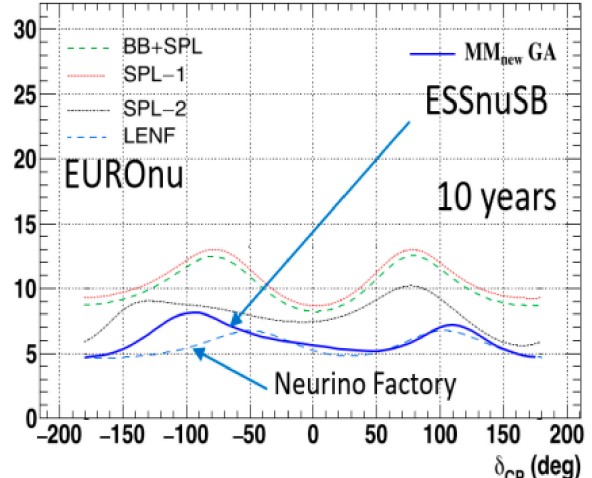

Error in $\delta_{CP}$ angle vs $\delta_{CP}$ angle after 10 years with 5% normalization error

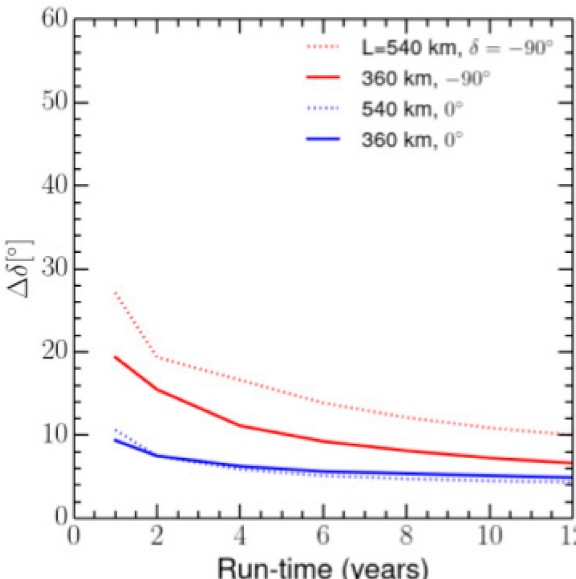

Error in $\delta_{CP}$ angle vs run time with 5% normalization error

**Figure 2.** Diagrams showing to the top left the ESSνSB performance for leptonic CP violation discovery in terms of number of signal standard deviations for different values of the CP violating phase angle $\delta_{CP}$ in degrees, in top right the precision in degrees with which $\delta_{CP}$ can be measured as function of $\delta_{CP}$, and at the bottom the same quantity as function of experimental run time in years. In all three plots, a systematic error of 5% has been conservatively assumed.

In addition to the leading measurement of leptonic CP violation, ESSνSB will measure the mixing angles of the Pontecorvo–Maki–Nakagawa–Sakata neutrino mixing matrix, in particular with the aim of solving the outstanding so-called octant problem, which is whether the value of mixing angle $\theta_{23}$ between the neutrino mass states two and three lies above or below $45°$. The large far ESSνSB neutrino detector will have the world's largest high-resolution water Cherenkov detector mass, more than twice that of Hyper-K, and therefore be a leading detector in the search for proton decay, reaching well above a proton lifetime of $10^{35}$ years. The ESSνSB neutrino detector will also be used as a high-performance neutrino observatory for measuring the neutrino pulses from future supernova explosions and the relic neutrinos remaining from all previous supernova explosions, as well as being a sensitive detector for the measurement of solar and atmospheric neutrinos.

**Funding:** This project was funded by the European Union's Horizon 2020 research and innovation programme under grant agreement No. 777419.

**Institutional Review Board Statement:** Not applicable.

**Informed Consent Statement:** Not applicable.

**Data Availability Statement:** The data presented in this study are available at https://essnusb.eu/DocDB/0013/001364/015/ESSnuSBCDR.pdf (accessed on 1 August 2022).

**Conflicts of Interest:** The author declares no conflict of interest.

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
