# Peer review of "ESS Neutrino Super Beam ESSνSB Design and Performance for Precision Measurements of the Leptonic CP Violating Phase δCP†"

_psf, doi:10.3390/psf8010069_

Round 1

Reviewer 1 Report

Comments and Suggestions for Authors

The submitted manuscript consists on the proceeding article of a conference talk, given at the NuFact conference in July 2022. The paper summarizes the main characteristics and physics sensitivity of the proposed oscillation neutrino facility ESSnuSB, a long-baseline experiment with a neutrino beam produced from pion decay-in-flight and observed using a Water Cherenkov detector. The peculiarity of this facility with respect to other proposals is the distance to the detector, which has been chosen to match the second oscillation maximum (instead of the first). As is nicely explained in the proceeding, observing oscillations at the second maximum is a priori better suited to observe CP violation; the price to pay is a consequent loss in statistics since the detector is located further away.

The proceeding summarizes nicely the features of the experiment and the main sensitivities expected. The only comment I have concerns the validity of some of the theory-based statements made by the author. I understand that this is the proceeding of a talk, and that the author is not a theorist, but nonetheless these statements should be either corrected (or better phrased) to avoid misinterpretation. Specifically, in the introduction the author claims that a measurement of leptonic CP violation is a ``primary milestone towards our understanding of dark matter'' and ``the expansion of the Universe'' (lines 28 and 33 in page 1, respectively). I am not sure what the author means by this, but as is written I believe that this is not correct. The discovery of leptonic CP violation may be related to the matter-antimatter asymmetry of the Universe and could have consequences for model building in relation to the neutrino mixing pattern, but I do not see a connection to dark matter or the expansion of the Universe. Similarly, in line 19 in page 2 the author writes ``Due to the oscillation between the different neutrino mass states,...''. I believe this is somewhat misleading, as the mass eigenstates do not oscillate (the flavor states do).

Once these statements have been corrected (or justified), in my opinion the manuscript will be ready for publication.

Author Response

I agree with the referee's comments.

In the line 33 the words "as well as what is the nature of the dark matter present in the universe" have been erased and in lines 37-38 the words "in particular on the nature of dark matter present in the universe and the expansion of the universe" have been erased.

In the lines 66-67 the "phrase "Due to the oscillation between the different neutrino mass states..." has been changed to "Due to the interference between the different neutrino mass states...."

I have also made some minor adjustments of my own  in the text here attached,

Tord Ekelof
